# A Novel Splice-Site Deletion in the *POU1F1* Gene Causes Combined Pituitary Hormone Deficiency in Multiple Sudanese Pedigrees

**DOI:** 10.3390/genes13040657

**Published:** 2022-04-08

**Authors:** Samar S. Hassan, Mohamed Abdullah, Katarina Trebusak Podkrajsek, Salwa Musa, Areej Ibrahim, Omer Babiker, Jernej Kovac, Tadej Battelino, Magdalena Avbelj Stefanija

**Affiliations:** 1Department of Pediatric Endocrinology and Diabetes, Gaafar Ibn Auf Pediatric Tertiary Hospital, Khartoum 11114, Sudan; mohamedabdullah@scda-sudan.org (M.A.); salwamusa@scda-sudan.org (S.M.); 2Sudan Childhood Diabetes Center, Khartoum 11111, Sudan; areejibrahim@scda-sudan.org (A.I.); omerbabiker@scda-sudan.org (O.B.); 3Department of Pediatrics, Faculty of Medicine, University of Khartoum, Khartoum 11115, Sudan; 4Clinical Institute of Special Laboratory Diagnostics, University Children’s Hospital, Ljubljana University Medical Centre, 1000 Ljubljana, Slovenia; katarina.trebusakpodkrajsek@mf.uni-lj.si (K.T.P.); jernej.kovac@kclj.si (J.K.); 5Faculty of Medicine, University of Ljubljana, 1000 Ljubljana, Slovenia; tadej.battelino@mf.uni-lj.si (T.B.); magdalena.avbelj@mf.uni-lj.si (M.A.S.); 6Department of Pediatrics and Child Health Faculty of Medicine, Al-Neelain University, Khartoum 11121, Sudan; 7Department of Paediatric Endocrinology, Diabetes and Metabolism, University Children’s Hospital, Ljubljana University Medical Centre, 1000 Ljubljana, Slovenia

**Keywords:** combined pituitary hormone deficiency, hypopituitarism, *POU1F1* gene, Dyke–Davidoff–Masson syndrome, meningitis

## Abstract

Pathogenic variants within the gene encoding the pituitary-specific transcription factor, POU class 1 homeobox 1 (*POU1F1*), are associated with combined pituitary hormone deficiency (CPHD), including growth hormone, prolactin, and thyrotropin stimulating hormone deficiencies. The aim of the study was to identify genetic aetiology in 10 subjects with CPHD from four consanguineous Sudanese families. Medical history, as well as hormonal and radiological information, was obtained from participants’ medical records. Targeted genetic analysis of the *POU1F1* gene was performed in two pedigrees with a typical combination of pituitary deficiencies, using Sanger sequencing, and whole-exome sequencing was performed in the other two pedigrees, where hypocortisolism and additional neurologic phenotypes were also initially diagnosed. In *POU1F1* gene (NM_001122757.2) a novel homozygous splice-site deletion—namely, c.744-5_749del—was identified in all 10 tested affected family members as a cause of CPHD. Apart from typical pituitary hormonal deficiencies, most patients had delayed but spontaneous puberty; however, one female had precocious puberty. Severe post-meningitis neurologic impairment was observed in three patients, of whom two siblings had Dyke–Davidoff–Masson syndrome, and an additional distantly related patient suffered from cerebral infarction. Our report adds to the previously reported *POU1F1* gene variants causing CPHD and emphasises the importance of genetic testing in countries with high rates of consanguineous marriage such as Sudan. Genetic diagnostics elucidated that the aetiologies of hypopituitarism and brain abnormalities, identified in a subset of affected members, were separate. Additionally, as central hypocortisolism is not characteristic of POU1F1 deficiency, hydrocortisone replacement therapy could be discontinued. Elucidation of a genetic cause, therefore, contributed to the more rational clinical management of hypopituitarism in affected family members.

## 1. Introduction

Congenital hypopituitarism (CH) is defined by the deficiency of one or more pituitary hormones and can be present in isolation or as part of a syndrome with associated midline defects of the forebrain or other associated phenotypes. Deficiency of at least two pituitary hormones, defined as the combined pituitary hormone deficiency (CPHD), occurs with the incidence of 1:4000 births [1]. Only 5–30% of cases with CPHD are familial, indicating a genetically inherited disorder [2]. There is a high clinical and genetic heterogeneity observed in CPHD, as disease-causing variants in more than 30 genes are associated with CPHD in humans [1]. The worldwide pathogenic variant frequency of the five most commonly affected genes (*POU1F1*, *PROP1*, *HESX1*, *LHX3*, *LHX4*) is 12.4%, which ranges from 11.2% in sporadic to 63% in familial cases with CPHD [3]. Nevertheless, *POU1F1* gene variants are a rare cause of CPHD, with frequencies as low as 0.5% in a large paediatric cohort of CPHD included in the GeNeSIS study [4], and up to 7.8% in another study that included 129 affected children [5].

The *POU1F1* gene (OMIM 173110) encodes POU class 1 homeobox 1, a transcription factor necessary for normal cell lineage specification of the anterior pituitary [6,7]. In 1992, four independent groups identified the *POU1F1* gene as the first identified genetic cause of CPHD [8,9,10,11]. It contains a transactivation domain and two homeobox domains, POU-specific and POU-homeo [12], with a high affinity to bind the promoters of genes encoding growth hormone (GH), prolactin (PRL), and thyroid-stimulating hormone (TSH) β subunit, binding sites being found also at the growth hormone-releasing hormone (GHRHR) gene promoter [13]. Disease-causing variants result in dominant or recessive GH and PRL deficiencies that are usually early and profound, with variable TSH deficiencies [1].

In this study, we describe four co-sanguineous families with CPHD associated with a novel *POU1F1* gene homozygous splice-site deletion.

## 2. Materials and Methods

Four partly interrelated co-sanguineous Sudanese families (Figure 1) with CPHD were tested for genetic aetiology. Only 10 out 11 subjects affected with CPHD were genetically tested, as subject FIV/21 died before the onset of this study, and his DNA sample was never obtained. Nevertheless, the clinical data of all 11 subjects were included in the study.

### 2.1. Anthropometry and Bone Age Estimation

Retrospective clinical details obtained from patients’ medical records included birth details, perinatal complications, history of consanguinity, family history, and parental heights. Height standard deviation scores (SD) were obtained using the 2000 CDC growth references. Bone age (BA) was estimated using the Greulich and Pyle method. The pituitary function was assessed using standard dynamic tests. Hormonal assays were performed using several commercial radioimmunoassay (RIA) kits, and age- and sex-related normal values for each hormone level were considered.

### 2.2. Hormonal Testing

GH was tested by one or two provocative tests in each patient, as indicated (insulin tolerance test and clonidine stimulation test). GH was measured by radioimmunoassay (RIA); levels less than 10 ng/mL were considered growth hormone deficiency (GHD). Thyroid hormone deficiency was tested by measuring thyroid-stimulating hormone (TSH) and free thyroxin (fT4) levels, using RIA. Diagnosis of central hypothyroidism was made if serum fT4 concentration was subnormal, with an inappropriately normal or low serum TSH concentration. Cortisol response was also determined during an insulin tolerance test. In the presence of hypoglycaemia (<2.6 mmol/L), the failure of the cortisol levels to increase twofold, compared with the baseline, and/or levels below 500 nmol/L (18.1 μg/dL) were considered hypocortisolism. PRL deficiency was diagnosed based on basal PRL levels below 3 ng/mL. The hypothalamic–pituitary–gonadal axis was assessed only when indicated by ICMA measuring basal luteinising hormone (LH) and/or stimulated LH in 45 min blood sample after 100 mcg of triptorelin given subcutaneously.

### 2.3. Magnetic Resonance Imaging (MRI)

Imaging included T1 and T2 weighted high-resolution pituitary imaging through the hypothalamic–pituitary axis (T1 sagittal 3 mm slices, T1 and T2 coronal 3 mm slices). Two neuroradiologists interpreted images, and details noted included the size of the anterior pituitary, position of the posterior pituitary signal, presence and morphology of the optic nerves, optic chiasm, pituitary stalk, septum pellucidum, and corpus callosum. Any additional brain anomalies were also assessed.

### 2.4. Genetic Analysis of The POU1F1 Gene 

DNA was extracted in all subjects from 5 mL of EDTA blood according to the established laboratory protocol using FlexiGene DNA Kit 250 (QIAGEN, Hilden, Germany). Targeted screening of the *POU1F1* gene by Sanger sequencing was performed in families I and II, with high clinical suspicion of *POU1F1* deficiency. The coding regions and the intron–exon borders were multiplied by polymerase chain reaction (PCR), with the designed oligonucleotide primers available on request. Sequencing was performed using a commercial kit BigDye Terminator v.3.1 Cycle Sequencing Kit and a 3500 Genetic Analyzer capillary electrophoresis system (Life Technologies, Foster City, CA, USA).

### 2.5. Whole-Exome Sequencing (WES)

Whole-exome sequencing (WES) as first-tier genetic testing was performed in selected members of families III and IV due to their complex phenotypes. DNA was isolated using the FlexiGene DNA Kit (QIAGEN) and stored at 4 °C until the NGS library preparation. Selected DNA samples from affected individuals were prepared for whole-exome sequencing using the Illumina DNA Prep with Enrichment Kit (Illumina, San Diego, CA, USA), according to the manufacturer’s protocol, and Agilent SureSelect V5 probes were used to enrich regions of the human exome. Libraries were PE150 sequenced using NovaSeq 6000 (Illumina), and sequencing data were analysed using the bcbio_nextgen bioinformatics pipeline (version 1.2.7; GRCh37 genome) [14]. The resulting VCF files were annotated, and candidate variants were filtered using VarAFT—a variant annotation and filtering tool [15] based on GnomAD frequency, reported ClinVar variants, and the CADD score prediction tool. Sanger sequencing was used to confirm familial co-segregation of the pathogenic variant.

## 3. Results

Eleven patients with hypopituitarism from four different families were included. Families I and II were interrelated, as were families III and IV, as shown in Figure 1. The deceased siblings of the family I were not assessed for their thyroid function, nor were they tested for GH deficiency. Their DNA samples were never taken. A male who died at the age of 3 years from severe pneumonia and sepsis had a history of hypotonia and developmental delay. The other deceased male sibling died at the age of one and a half years, following fever and convulsion due to severe malaria. The two female siblings died at 4 and 5 months of age at home, with severe untreated gastroenteritis. All patients were born at term, with a mean birth weight of 2500 g. In Sudan, the cutoff values for low birth weight (10th percentile = 2400 gm) are lower than the international cut-off values [16].

### 3.1. The Phenotype of Patients with a Novel Variant in the POU1F1 Gene

All affected patients manifested with profound GH, TSH, and PRL deficiency. Their clinical characteristics are summarised in Table 1, while their hormonal values are presented in Table 2, and their brain MRI results are presented in Table 3. TSH deficiency (TSHD) was the first presenting hormonal deficiency in all affected siblings, and all were diagnosed within the first 3 months of life, with a mean age of diagnosis of 29 days (Table 1). Symptoms included prolonged neonatal jaundice, coarse cry, feeding difficulties, dry skin, constipation, and failure to thrive. None of them had any episodes of hypoglycaemia or midline septal defects, and all affected males had normal penile lengths. Diagnosis of hypothyroidism was late due to the lack of newborn screening programs in our setting. All patients were treated with L-thyroxin in the context of congenital hypothyroidism, but none was believed to have hypopituitarism until later when they presented with short stature. Generally, referral for short stature was late, with a mean age at presentation of 5 years, and the mean height standard deviation (SD) from the mean was −4.01 for males and −2.9 for females (Table 1). All affected patients had characteristic midfacial hypoplasia, depressed nasal bridge, and high-pitched voice. GH supplementation was prescribed to all patients; however, two severely neurologically debilitated siblings (FIV/21 and FIV/22) discontinued therapy within 1 year due to worsening kyphoscoliosis. Following the initiation of GH therapy, all affected patients showed a response in growth; this was evaluated by growth velocity every six months and annual bone ages. Insulin growth factor one (IGF-1) was not available in our settings to use for follow-up of GH therapy. Most patients that achieved final heights were short (Table 1). All patients with closed growth plates and a growth velocity of less than 2 cm/year were transitioned to adult services for the continuation of adult dose GH therapy.

The majority of affected members attained puberty late but spontaneously; the mean age at onset of puberty in our study for males was 14 and for females, 12 years, and the mean age at menarche was 14.6 years. Exceptionally, a girl FIII/14 had precocious puberty, as determined by menarche at 9 years and 2 months. Additionally, one male, FIII/16, received sex steroids for delayed puberty, but normal gonadotropins (LH levels of 22.7 mIU/mL after LHRH analogue stimulation) and testosterone levels (403 ng/dL (normal range 300–1200)), along with testicular growth (12 mL bilaterally), were identified at age 18 years.

Patients FIII/14, FIII/16, and FIV/22 complained of nocturnal enuresis. Diabetes insipidus was excluded by normal urine outputs per day (≤2 L/24 h), normal values of urine specific gravity (1.015–1.030), serum sodium (135–145 mmol/L), and serum osmolarity of (285–295 mOSM/kg), and all had a normal bright spot of the posterior pituitary with intact stalks on brain MRI images. A water deprivation test was not indicated, and desmopressin (Minirin melt) tabs 120 mg OD was initiated for treatment of secondary enuresis with a good response. At the time of publication, only patients FIII/14 and FIV/22 were still on desmopressin.

All affected siblings of families III and IV at the time of diagnosis of hypopituitarism initially showed sufficient cortisol responses to the insulin tolerance test (ITT) (Table 1); then at some point during their years of follow-up, hydrocortisone was empirically commenced when they presented with symptoms of fatigue and borderline morning cortisol levels (6–17 µg/dL) (Table 1). They received 10 mg/SA/day of hydrocortisone for a mean period of 9 years. Following the identification of a *POU1F1* variant, all mentioned cases were successfully weaned off hydrocortisone replacement. Re-evaluation of the pituitary–adrenal axis was performed by serial monthly morning cortisol levels. Levels 18 µg/dL or more were considered normal and indicated recovery of the hypothalamic–pituitary–adrenal axis. Synacthen was not available in our settings, but all patients were given alert cards and educated on stress dosing if needed. Four months after stopping hydrocortisone in patients FIII/14, FIII/16, FIII/17, and FIV/22, an ITT was performed to evaluate cortisol sufficiency. Results revealed suboptimal cortisol responses, the highest level of which was 12.1 ng/dL (Table 4).

The affected siblings of family II had significant learning disabilities and behavioural problems. Patients FIII/14, FIV/21, and FIV/22 were diagnosed with meningoencephalitis at ages 34 months, 4 months, and 2 years, respectively. The diagnosis was made according to classical clinical presentation; however, cerebral spinal fluid samples were not obtained. Despite receiving broad-spectrum antibiotics, they all recovered with permanent neurological deficits. Patient FIII/14 ultimately had spastic hemiplegia and global developmental delay (gross and fine motor, speech, and social), and brain MRI showed a large left cerebral post-infective infarction. Both siblings, FIV/21and FIV/22, ultimately had spasticity and global developmental delay. Patient FIV/21 was completely bedridden, while his sibling’s (FIV/22) movement was limited to crawling, thus requiring a wheelchair for movement, she also had aggressive behaviour. Their brain MRI showed unilateral cerebral hemiatrophy, with marked loss of white, subcortical, and deep matter volumes with prominent sulci, secondary ballooning of the ipsilateral lateral ventricle, and no midline shifting, likely as a consequence of a postnatal cerebral infarction (Figure 2). All three patients continued to have recurrent seizures, which were difficult to control despite multiple anticonvulsant therapies. At the age of 17 years, before the initiation of this study, patient FIV/21 suddenly passed away at home, with no specific events before death. Pituitary imaging performed on other patients demonstrated a normal posterior bright spot and intact stalk. While the anterior pituitary was mostly hypoplastic, as shown in Table 3, empty sella turcica was observed in one patient.

### 3.2. Patients’ Genotype Details

A novel homozygous 11 base pair deletion, *POU1F1* (NM_001122757.2):c.744-5_749del11bp, was identified in all affected family members tested. The deletion crossed the exon–intron junction at the 5′ end of exon 6, as illustrated in Figure 3, along with the schematic location of previously published *POU1F1* variants implicated in CPHD. According to the GnomAD, it has not been reported in the general population [17], and according to the ACMG criteria, the variant was classified as pathogenic (PVS1, PM2, PP3). The variant carrier status of their unaffected parents and siblings is shown in Figure 1. 

Additional genetic variants were searched for in the living patients of families III and IV for potentially increased susceptibility to ischemic brain insults. In the patient FIV/22, a novel heterozygous missense variant in the gene encoding fibrinogen β chain *FGB* (NM_001184741.1):c.403A>C (p.N135H). The variant has not been reported in the general population, as reported by GnomAD. It was classified, according to the ACMG criteria (PM2, PP2, PP3), as a variant of unknown significance. The variant was absent in the unaffected siblings of family IV; however, the DNA sample of the additional affected sibling FIV/21 was not available due to his premature death. In family III, the variant was also detected in three siblings, one of them homozygous, but it was not detected in subject FIII/14, who suffered from brain infarction.

## 4. Discussion

The results of the genetic analysis of four inter-related and co-sanguineous pedigrees with multiple relatives having CPHD expand the spectrum of pathogenic variants implicated in CPHD, with a novel splice-site deletion in the *POU1F1* gene. To date, more than 40 variants in the *POU1F1* gene associated with hypopituitarism are reported, most located within the POU-specific and POU-homeodomains, as summarised in Figure 2 [4,5,8,9,10,11,18,19,20,21,22,23,24,25,26,27,28,29,30,31,32,33,34,35,36,37,38,39,40,41,42], including large deletions encompassing part of the gene or entire coding region [10,43,44,45]. Most variants are inherited recessively and result in severe and early onset GH, TSH, and PRL deficiencies. Some missense variants and variants affecting exon 2 skipping exert a dominant-negative effect [9,11,26,35,40,41], and a dosage effect was observed in a family carrying a truncating variant at the level of the POU-specific domain [30]. Genotype–phenotype correlation was observed in most families, including the p.E230K and p.F135C variants related to later-onset central hypothyroidism [5,18,46], the p.P76L related to isolated GHD [20], or the c.214+1G>T related to mild combined GHD and PRL deficiency [40].

The variant identified in our pedigree was expected to be pathogenic, as it affected the exon-splicing site of exon 6, which encodes for the major part of the POU-homeodomain (Figure 2). POU homeodomain is necessary for DNA binding of the POU1F1 transcription factor [20]. Previously reported frameshift mutations leading to truncation of the POU homeodomain, c.605delC, c.747delA, and c.775dupA resulted in vitro in a loss of DNA binding and decreased transactivation of *GH1*, *PRL*, and *POU1F1* promoters [5,24,31]. Patients with previously reported POU-homeodomain-truncating variants clinically presented with autosomal recessive early-onset and severe TSH, GH, and PRL deficiencies [5,31,32,33], similar to the clinical course observed in our patients, which supports the pathogenic nature of the c.744-5_749del11bp variant. 

All here-reported cases presented with profound symptomatic TSHD during the first months of life, manifesting as prolonged neonatal jaundice and feeding difficulties. Nevertheless, affected siblings of families II and III had a rather delayed diagnosis of TSHD and later presented with poor school performance and learning disabilities. Mental development in *POU1F1*-associated CPHD is usually not impaired, but patients with late or inadequate levothyroxine supplementation can have related neurodevelopmental consequences [39,47]. In our patient’s area of residence, there was no neonatal screening. Only very few countries in sub-Saharan Africa have some newborn screening [48,49,50,51], and only the TSH newborn screening method is adopted in the African region [48,49,50,51]. Worldwide, only a few countries screen newborns for central hypothyroidism, including The Netherlands, where central congenital hypothyroidism occurs with an incidence of 1:13,000 newborns [52]. Their experience shows that the diagnosis is often established only after neonatal screening results despite clinically symptomatic hypothyroidism [53], which indicates a challenging clinical diagnosis. This emphasises the importance of newborn screening using both TSH and free T4 levels, especially in countries with high rates of consanguineous marriage such as Sudan.

While GHD was confirmed in all of the patients, the peak GH values ranged from undetected values up to 2.1 ng/dL, demonstrating a slight variability in the GHD severity, which might be a feature of this *POU1F1* variant. The patients also had variable proportional short stature at presentation, and all presented with the unique facial characteristics described in the literature—infantile facial features, crowded teeth, and frontal bossing. While the characteristic facial dysmorphism could participate in the early diagnosis of this rare disorder, the delayed referral to endocrine services may be due to the unawareness of general physicians and paediatricians in our setting about the clinical presentation of hypopituitarism. It also may be due to the lack of families’ concern about short stature. The unsatisfactory adult heights of our patients were affected by multiple factors, particularly the delayed age at presentation and initiation of GH therapy, with the mean age at presentation of 5 years. Even though GH therapy was provided by the Sudanese government free of charge in our centre, there were problems with continuous availability throughout the year due to irregular follow-up visits and poor adherence to therapy, as well as an occasional shortage of GH from the suppliers. To assure a continuous supply of GH therapy, a three-month supply of GH was provided for patients residing far from our centre. Still, some families found it difficult to travel to Khartoum every three months. All reported cases in this study resided outside Khartoum, the capital of Sudan. In addition, follow-up of GH therapy in our settings depended on three-to-six months of growth velocity and annual wrist X-rays for bone age estimation. In our settings, IGF-1 measurements were highly expensive and unaffordable to families.

There is a spectrum of pubertal development reported in patients with *POU1F1* variants, ranging from central precocious puberty (CPP) identified in two unrelated subjects with CPHD [23] up to the absent puberty in a 17-year-old male having severe untreated TSH and GH deficiencies, in whom puberty began spontaneously after adequate replacement of thyroid and growth hormones [47]. The mean ages at onset of puberty in our study for males and females were 14 and 12 years, respectively, and the mean age at menarche was 14.6 years. All developed puberty spontaneously, except for patient FIII/16, who unnecessarily received sex steroids replacement up to genetic diagnosis. While the onset of puberty in our cohort was delayed, compared with worldwide populations [54], it was not delayed in girls, compared with the population of Zambia, where the onset of puberty was 11.5 years in urban girls and 13.2 years in rural girls, with the overall age at menarche being 14.8 years [55]. CPP, although reported in CPHD, is an extremely rare finding. No relationship has been established between *POU1F1* genotype and CPP in humans. In our patient FIII/14, who presented with menstrual cycles at the age of 9 years and 2 months, the triggering factor of her early puberty was most likely secondary to her underlying cerebral infarction, which resulted from an infective brain insult. Whether altered pubertal timing observed in our cohort was related to POU1F1 deficiency remains an open question, particularly since animal studies have shown that the Pou1f1 affects the regulation of genes encoding GnRH receptor and Gata2 consequently influencing cell differentiation and gene expression throughout the reproductive axis [56].

Affected patients in families III and IV were diagnosed to have secondary hypocortisolism based on suboptimal cortisol concentrations and symptoms of fatigue and lethargy and received hydrocortisone supplementation for a mean period of 9 years. The differentiation between hypocortisolism due to ACTH deficiency and secondary pituitary suppression from chronic use of steroids was very challenging in our setting, where synacthen is not available. With genetic testing results, hydrocortisone treatment in affected patients of family III and IV was stopped, along with instructions on stress dosing and alert cards provided. Four months later cortisol response to hypoglycaemia using ITT was still insufficient. Ongoing hypocortisolism could be a consequence of a long-term steroid replacement; nevertheless, the contribution of additional factors, such as a hypoplastic pituitary gland, could not be excluded. 

Both siblings affected with CPHD in family IV (FIV/21 and FIV/22) were diagnosed also with spasticity, seizures, mental retardation, and unilateral cerebral hemiatrophy on neuroimaging, which have been reported as Dyke–Davidoff–Masson syndrome (DDMS) [57]. Cerebral hemiatrophy is known in the literature as a result of brain insult which either occurs in utero or within the first three years of life [58]. Additional distantly related patient FIII/14 also suffered from ischemic brain insult, and in all three subjects, this was likely a complication of bacterial meningitis, which occurred within the first three years of age. MRI is a helpful tool in differentiating between prenatal and postnatal causes of brain insult. Insult occurring in utero results in the consequent displacement of the structures of the midline to the side of the pathology and is associated with a lack of sulcus prominences [58]. The MRI imaging results in our patients were characteristic of a postnatal brain insult. This devastating complication occurred in all three patients of our cohort who suffered from bacterial meningitis. In general, bacterial meningitis could cause brain ischemia in 10–29% of adult cases [59] and 3–14% of paediatric cases [60], the main mechanisms of which are inflammatory vasculopathy and intravascular thrombus formation [59]. Studies indicate a higher incidence of bacterial meningitis and related neurological sequelae in children from developing countries, due to poorer vaccination rates and delayed treatments [60]. Nevertheless, the question was whether our patients had any additional risk factors for ischemic brain insults.

A novel missense variant of uncertain significance in *FGB*, which encodes fibrinogen β chain, was identified in one of the two tested patients suffering from brain infarction. Upon cleavage by the protease thrombin, fibrinogen β polymerises with fibrinogen α and fibrinogen γ to form fibrin and thus plays an essential role in blood coagulation. Pathogenic dominant variants in *FGB* cause reduced levels of fibrinogen, altered fibrinogen activity, or both, which predisposes to either arterial/venous thromboses or a bleeding disorder [61]. Fibrinogen levels were normal in patient FIV/22, and there was no other clinical evidence of bleeding or clotting dysfunction in the affected patients or their parents. The lack of segregation of the detected *FGB* variant with the phenotype suggests it was most probably not associated with brain infarction in these families.

Identifying the genetic aetiology of CPHD in the four pedigrees facilitated personalised medical management of the affected individuals. They were spared from the further continuation of unnecessary hormonal testing and steroid and sex hormone replacement therapy. Genetic information thus helped reduce cost effects in management plans, particularly in the low economical setting, where the high cost of investigations is not covered by health insurance, nor is it affordable by the family. In addition, a multifactorial background of the complex clinical presentation was uncovered in families III and IV.

## 5. Conclusions

In conclusion, a novel *POU1F1* 11 bp deletion crossing a splice-site of exon 6, which encodes the POU homeodomain, was identified in a cohort of 11 patients with early-onset profound CPHD from four Sudanese families. A subset of patients had additional phenotypes of Dyke–Davidoff–Masson syndrome, which was aetiologically unrelated. We highlight the importance of genetic testing in populations with high rates of consanguineous marriage such as Sudan and its helpful role in proper phenotype characterisation in patients with CPHD. In the presented families, genetic diagnostics contributed to the more rational clinical management of affected family members, which is of value in a resource-limited environment.

## Figures and Tables

**Figure 1 genes-13-00657-f001:**
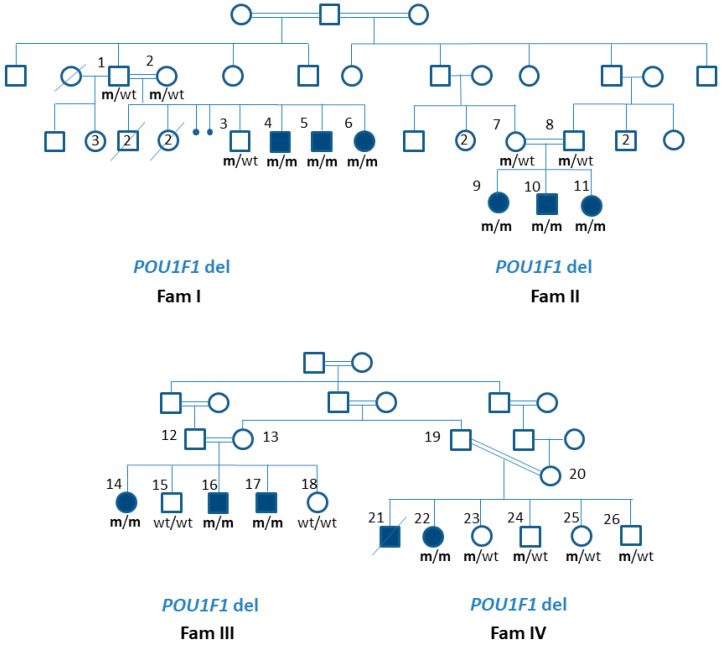
Pedigrees and gene variant status in families with combined pituitary hormone deficiencies.

**Figure 2 genes-13-00657-f002:**
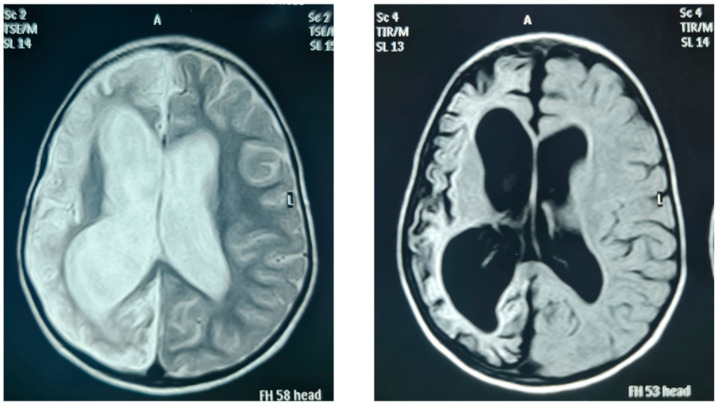
Brain MRI of patient FIV/22.

**Figure 3 genes-13-00657-f003:**
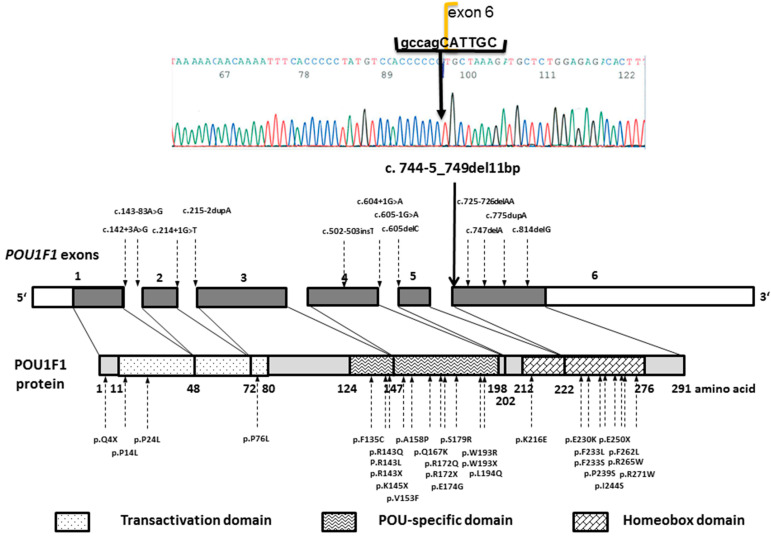
Schematic representation of *POU1F1* gene and protein with marked previously reported variants and the novel *POU1F1* deletion.

**Table 1 genes-13-00657-t001:** Selected clinical characteristics of patients with combined pituitary hormone deficiency: GH, growth hormone; GHD, growth hormone deficiency; M, male; F, female; MC, menstrual cycle; SD, standard deviation; NA, not available.

Family	Gender	Birth Weight (g)	Age at Diagnosis of Hypothyroidism (Days)	Age at Diagnosis of GHD (Years)	Height at Diagnosis of GHD (cm) (SD)	Bone Age in Years	Age at Start of Puberty (Years)/Menarche (m)	Mid Parenteral Height (cm)	End Adult Height (cm)	Outcome
FI/4	M	3000	9	10	123 (−2.5)	NA	15	172	153	Short adult height
FI/5	M	2800	10	7	103 (−3.5)	5	14	172	160	Short adult height
FI/6	F	3100	4	5	100 (−1.7)	4	13/MC = 16	159	144.5	Short adult height
FII/9	F	2800	90	10	106 (−5.6)	5.9	13/MC = 16	161	134	Short adult height
FII/10	M	3000	60	4	86 (−4.6)	1.7	15	174	Still on GH	Still on GH
FII/11	F	3000	32	2	74 (−3.4)	0.5	14/MC = 16	161	Still on GH	Still on GH
FIII/14	F	2500	38	8	103 (−5.2)	5	7/MC = 9.2	160.5	149	Short adult height
FIII/16	M	2000	20	4	97 (−1.2)	NA	14	173.5	165	Reached genetic target range
FIII/17	M	2000	10	1.5	67 (−4.8)	NA	13	173.5	Still growing	Still on GH
FIV/21	M	3000	10	10	93.5 (−7.5)	NA	NA	177	Deceased	Deceased
FIV/22	F	2800	40	4	85 (−3.85)	3	13/MC = 16	164	113	Short adult height
Mean for parameters		2450	29	5	M = −4.01 F = −2.9	-	M = 14.2 F = 12/MC = 14.6			-

**Table 2 genes-13-00657-t002:** Selected hormonal values in patients with CPHD: NA, not available; GH, growth hormone; TSH, thyroid-stimulating hormone; LH, luteinising hormone; FT4, thyroxine.

Family Number	Peak GH ng/mL (≥10)	Peak Cortisol µg/dL (≥18)	Am Cortisol µg/dL	TSH mIU/mL (0.2–4.9)	FT4 ng/dL	Prolactin ng/mL	Basal LH mIU/mL (≥0.3)
FI/4	˂0.07	22.4	14.7	<0.02	1.2 (1.1–1.5)	0.9	5.9
FI/5	˂0.07	18.1	7.3	2.4	0.9 (1.1–1.6)	0.9	2.7
FI/6	˂0.07	16.1	14.8	<0.02	1.5 (1.1–1.35)	0.9	4.1
FII/9	˂0.07	34.5	NA	<0.01	4.1 (4.9–11)	0.6	1.7
FII/10	N/A	21.2	NA	<0.005	3.2 (5.1–14.1)	0.8	2.5
FII/11	N/A	19.9	NA	<0.005	7.2 (6.5–13.3)	0.7	1.9
FIII/14	2.1	13	15.7	<0.02	2 (7.0–17)	˂1.0	1.13
FIII/16	2.1	15.5	NA	0.1	1.7 (7.0–17.0)	˂1.0	NA
FIII/17	0.05	24.9	11.4	0.3	6.6 (3.8–11.2)	NA	NA
FIV/21	0.31	NA	7	<0.02	NA	0.1	NA
FIV/22	0.36	NA	15.6	<0.02	2.1 (2.9–5.1)	0.1	22.1

**Table 3 genes-13-00657-t003:** Results of the MRI imaging: NA, not available.

Family Number	MRI Findings
FI/4	Hypoplastic anterior pituitary gland, normal posterior gland, and intact stalk
FI/5	Hypoplastic anterior gland, normal posterior gland, and intact stalk
FI/6	NA
FII/9	Empty sella turcica
FII/10	NA
FII/11	NA
FIII/14	Large left cerebral infarction with normal pituitary gland
FIII/16	NA
FIII/17	Hypoplastic anterior gland
FIV/21	Unilateral cerebral diffuse cortical gliosis, atrophy with evacuee dilatation of the ipsilateral ventricle, prominent sulci, and hypoplastic anterior pituitary gland
FIV/22	Right cerebral hemiatrophy, with marked loss of white, subcortical, and deep matter volumes with prominent sulci, and secondary ballooning of the right lateral ventricle with no midline shifting, right temporal cystic encephalomalacia

**Table 4 genes-13-00657-t004:** Cortisol responses based on insulin tolerance test in patients from families III and IV after cessation of hydrocortisone.

Index Case	Blood Glucose mg/dL (mmo/L)	Cortisol ng/mL (mmol/L)
FIII/14	32 (1.8)	12.1 (333.8)
FIII/16	36 (2.0)	6.1 (168.2)
FIII/17	45 (2.5)	10.7 (295.2)
FIV/22	40 (2.2)	7.0 (193.1)

## Data Availability

The data presented in this study are available on request from the corresponding author. The data are not publicly available due to the privacy restrictions.

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
