# Peer review of "A Novel Splice-Site Deletion in the POU1F1 Gene Causes Combined Pituitary Hormone Deficiency in Multiple Sudanese Pedigrees"

_genes, 2022, doi:10.3390/genes13040657_

Round 1

Reviewer 1 Report

The study follows the pedigrees of four Sudanese families with combined pituitary hormone deficiency (CPHD).

The authors performed targeted screening of the POU1F1 gene by Sanger sequencing in families I and II. Whole exome sequencing was performed in selected family members of families II and IV, after which Sanger sequencing was used to confirm familial cosegregation of the pathogenic variant.

My main comment is why the same genotyping procedure was not carried out for all four families?

It would be very interesting to know the genotypes of the deceased two male and two female children in the family 1 and the reason of their death. It would also be good to know the genotypes of two miscarriages in the same family.

In some members of families III and IV, due to the sequencing of the whole exome, a missense variant of the gene encoding fibrinogen beta chain (FBC) was detected, and therefore the same methodology is necessary in families I and II.

Author Response

Response to Reviewer 1 Comments

Point 1: The study follows the pedigrees of four Sudanese families with combined pituitary hormone deficiency (CPHD). The authors performed targeted screening of the POU1F1 gene by Sanger sequencing in families I and II. Whole exome sequencing was performed in selected family members of families II and IV, after which Sanger sequencing was used to confirm familial cosegregation of the pathogenic variant.

Authors’ response: We thank the reviewer 1 for the insightful and detailed review of our manuscript. A detailed response to each of the comment was submitted individually. The suggestions of the reviewers were incorporated into the revised version of the manuscript and were highlighted in yellow.

Point 2: My main comment is why the same genotyping procedure was not carried out for all four families?

Authors’ response: Thank you for this important remark. In families 1 and 2 we decided to employ a Sanger sequencing genetic approach because of the typical phenotype suggesting  the POU1F1 deficiency related CPHD. Nevertheless, in the case of the negative Sanger sequencing results in these two families, it was planned to continue with the WES analysis as a second tier testing.

In families III and IV, clinical phenotypes were more complex and therefore the WES was issued at the very beginning.

To elucidate this, we have rewritten the following sentences in the Materials and Methods section now reading:

Targeted screening of the POU1F1 gene by Sanger sequencing was performed in families I and II with high clinical suspicion of POU1F deficiency.

Whole exome sequencing (WES) as a first tier genetic testing was performed in selected members of families III and IV due to their complex phenotypes.

Point 3: It would be very interesting to know the genotypes of the deceased two male and two female children in the family 1 and the reason of their death. It would also be good to know the genotypes of two miscarriages in the same family.

Authors’ response: We agree that knowing the genotype of the deceased family member /miscarriages would be of high interest. Unfortunately, the DNA samples of these family members are not available. We agree indeed that revealing the causes of death in deceased siblings of family 1 is interesting for readers , hence we added to result section the following marked in yellow in the main manuscript:

The deceased siblings of family 1 were not assessed for their thyroid function, nor were they tested for GH deficiency. Also their DNA samples were never taken. A male who died at the age of 3 years for severe pneumonia and sepsis had a history of hypotonia and developmental delay. The other deceased male sibling died at the age of one and a half years following fever and convulsion due to severe malaria.  The two female siblings died at 4 and 5 months of age at home with severe untreated gastroenteritis.

Point 4: In some members of families III and IV, due to the sequencing of the whole exome, a missense variant of the gene encoding fibrinogen beta chain (FBC) was detected, and therefore the same methodology is necessary in families I and II.

Authors’ response: Indeed, as stated in the Results section novel FGB variant of uncertain significance was detected in the patient FIV/22 with brain infarction. In family III, the variant was also detected in three siblings, one of them being homozygous, but it was not detected in the subject FIII/14, who suffered from brain infarction. In addition, fibrinogen levels were normal in the patient FIV/22 and there was no other clinical evidence of bleeding or clotting dysfunction nor in the affected patients or their parents. Therefore, the detected FGB variant is most probably not associated with the brain infarction in these families.

Since brain infarction was not reported in the families I and II and the genetic cause of the brain infarction in related families III and IV was not identified, it was not reasonable to perform additional genetic testing in families I and II.

Nevertheless, we would like to apologise for the unfortunate mistake regarding the FGB variant in the Discussion section. In the results it was correctly stated, that one patient (FIV/22) with brain infarction had novel FGB variant, but the other (FIII/14) did not.   However, in the Discussion it was stated that both patients had this variant present. Therefore, we had corrected this and is now reading:

A novel missense variant of uncertain significance in FGB, which encodes fibrinogen beta chain, was identified in one of the two tested patients suffering from brain infarction.

In addition to clarify the impact of this variant additional sentence was added to the Discussion section reading:

The lack of segregation of the detected FGB variant with the phenotype suggests it was most probably not associated with the brain infarction in these families.

Reviewer 2 Report

This is a well-written paper on a complex topic. Hassan et al. reported that a novel homozygous splice site deletion in the POU1F1 gene was identified as the causative genetic abnormality in 4 families 10 cases with CPHD. They used WES as a tool to identify the mutation, which is currently widely used to diagnose patients with congenital anomalies. In addition, the clinical characteristics of patients with CPHD in 4 families 11 cases were summarized.

1) Page 5, in the endocrine assessment section. In the patient with POU1F1 mutation, it is known to be variable from early central congenital hypothyroidism to the maintenance of thyroid function in adulthood, although GH and PRL deficiencies are usually present. However, in 2 of 11 cases, peak GH was 2.1 ng/dL in the response to insulin hypoglycemic stimulation. Variability of GH deficiency might be a feature of this POU1F1 mutation?

2) Page 5, in endocrine assessment section onward. The cortisol to insulin hypoglycemic stimulation was less than 18 µg/dL in 3 of 11 cases with the POU1F1 mutation. In addition, some cases had hypoadrenocorticism with actual malaise and lethargy. Had the hypoplastic pituitary gland influenced the corticotrophs, although the developmental process is different from POU1F1 lineage cells?

3) Page 9, in the discussion section. It may be difficult to provide a mechanism for the late-onset or early onset of puberty in this case, but it is discussed carefully. Although it is not fully clear in humans, animal studies have shown that the Pou1f1 gene affects GnRH receptor function and regulation of the Gata2 gene, which may be included in the discussion.

Author Response

Response to Reviewer 2 Comments 

Point 1: This is a well-written paper on a complex topic. Hassan et al. reported that a novel homozygous splice site deletion in the POU1F1 gene was identified as the causative genetic abnormality in 4 families 10 cases with CPHD. They used WES as a tool to identify the mutation, which is currently widely used to diagnose patients with congenital anomalies. In addition, the clinical characteristics of patients with CPHD in 4 families 11 cases were summarized.

Authors’ response: We thank the reviewer 2 for the insightful and detailed review of our manuscript. A detailed response to each of the comment was submitted individually. The suggestions of all reviewers were incorporated into the revised version of the manuscript and were highlighted in yellow.

Point 2: Page 5, in the endocrine assessment section. In the patient with POU1F1 mutation, it is known to be variable from early central congenital hypothyroidism to the maintenance of thyroid function in adulthood, although GH and PRL deficiencies are usually present. However, in 2 of 11 cases, peak GH was 2.1 ng/dL in the response to insulin hypoglycemic stimulation. Variability of GH deficiency might be a feature of this POU1F1 mutation?

Authors’ response: Thank you for this interesting observation and suggestion. We added  to the discussion section the following sentence highlighted in yellow in the manuscript to present this possibility:

While GHD was confirmed in all the patients, the peak GH values ranged from undetected values up to 2.1 ng/dL, demonstrating a slight variability in the GHD severity, which might be a feature of this POU1F1 variant. The patients also had variably severe proportional short stature at presentation…

Point 3: Page 5, in endocrine assessment section onward. The cortisol to insulin hypoglycemic stimulation was less than 18 µg/dL in 3 of 11 cases with the POU1F1 mutation. In addition, some cases had hypoadrenocorticism with actual malaise and lethargy. Had the hypoplastic pituitary gland influenced the corticotrophs, although the developmental process is different from POU1F1 lineage cells?

Authors’ response: Thank you for this remark. Indeed we agree that cut off levels for cortisol response to hypoglycemia is 18µg/dl.  Indeed, we expected the resolution of hypocorticism after cessation of glucocorticoid replacement. Perhaps the follow-up was too short. A synacthen stimulation test unfortunately is not available in our settings. Nevertheless, at the moment additional factors participating to ongoing hypocorticism cannot be excluded.  We added the following statements in the Discussion section ( highlighted in yellow in the manuscript)

Ongoing hypocortisolism could be a consequence of a long-term steroid replacement; yet, the contribution of additional factors, such as a hypoplastic pituitary gland, could not be excluded.

 Point 4:  Page 9, in the discussion section. It may be difficult to provide a mechanism for the late-onset or early onset of puberty in this case, but it is discussed carefully. Although it is not fully clear in humans, animal studies have shown that the Pou1f1 gene affects GnRH receptor function and regulation of the Gata2 gene, which may be included in the discussion.

Authors’ response: Thank you for this important remark and insight in this additional published data, which suggests possible relation of altered pubertal timing with POU1F1 deficiency. To expose this possibility, we have added the following sentence to the Discussion section  (highlighted in yellow in the main manuscript):

Whether altered pubertal timing in our cohort was related to POU1F1 deficiency remains an opened question, particularly since animal studies have shown that the Pou1f1 affects regulation of genes encoding GnRH receptor and Gata2 consequently influencing cell differentiation and gene expression throughout the reproductive axis.

Reviewer 3 Report

This manuscript describes a study of the impact of mutations in the POU1F1 gene in 4 families with a high rate of consanguineous marriage.  The study focuses on the genotype and phenotype of these affected progeny and shows how the mutated gene can drastically impair growth, thyroid function, and development.  It is of value as it teaches the negative aspects of consanguineous marriage as well as the importance of early treatment and diagnoses.  The only concerns I have are minor

Table 1.  I believe that diseased should be deceased.

Table 2. should be sella turcica. 

The study is well illustrated and well presented.  

Author Response

Response to Reviewer 3 Comments

Point 1: This manuscript describes a study of the impact of mutations in the POU1F1 gene in 4 families with a high rate of consanguineous marriage.  The study focuses on the genotype and phenotype of these affected progeny and shows how the mutated gene can drastically impair growth, thyroid function, and development.  It is of value as it teaches the negative aspects of consanguineous marriage as well as the importance of early treatment and diagnoses.  The only concerns I have are minor. The study is well illustrated and well presented.  

Authors’ response: We thank the reviewer 3 for the insightful and detailed review of our manuscript. A detailed response to each of the comment was submitted individually. The suggestions of the all reviewers were incorporated into the revised version of the manuscript and were highlighted in yellow.

Point 2: Table 1.  I believe that diseased should be deceased.

Authors’ response: Thank you. We have corrected the misspelling.

Point 3: Table 3. should be sella turcica. 

Thank you. We have corrected the misspelling.

Round 2

Reviewer 1 Report

The manuscript has been improved with added comments. I'm satisfied with the revised version of the manuscript and have no further comments.

Author Response

we thank reviewer #1 for his valuable input and review .